# Relationship Between the Presence of Red Complex Species and the Distribution of Other Oral Bacteria, Including Major Periodontal Pathogens in Older Japanese Individuals

**DOI:** 10.3390/ijms252212243

**Published:** 2024-11-14

**Authors:** Mariko Kametani, Yasuyuki Nagasawa, Momoko Usuda, Ami Kaneki, Masashi Ogawa, Kensaku Shojima, Hiromitsu Yamazaki, Kana Tokumoto, Daiki Matsuoka, Kana Suehara, Yuto Suehiro, Tatsuya Akitomo, Chieko Mitsuhata, Taro Misaki, Seigo Ito, Shuhei Naka, Michiyo Matsumoto-Nakano, Kazuhiko Nakano, Hiromitsu Kishimoto, Ken Shinmura, Ryota Nomura

**Affiliations:** 1Department of Pediatric Dentistry, Hiroshima University Graduate School of Biomedical and Health Sciences, Hiroshima 734-8553, Japan; mrysk25@hiroshima-u.ac.jp (M.K.); musuda@hiroshima-u.ac.jp (M.U.); kaneki@hiroshima-u.ac.jp (A.K.); caries0@hiroshima-u.ac.jp (M.O.); takitomo@hiroshima-u.ac.jp (T.A.); chiekom@hiroshima-u.ac.jp (C.M.); 2Department of General Medicine, Hyogo Medical University, Nishinomiya 663-8501, Japan; kshojima@hyo-med.ac.jp (K.S.); hiromitu@hyo-med.ac.jp (H.Y.); shimmura@hotmail.co.jp (K.S.); 3Department of Dentistry and Oral Surgery, Hyogo Medical University, Nishinomiya 663-8501, Japan; ka-tokumoto@hyo-med.ac.jp (K.T.); kisihiro@hyo-med.ac.jp (H.K.); 4Department of Pediatric Dentistry, Okayama University Graduate School of Medicine, Dentistry and Pharmaceutical Sciences, Okayama 700-8558, Japan; kujiraking0718@s.okayama-u.ac.jp (D.M.);ph2s4pws@s.okayama-u.ac.jp (K.S.); ppvk0xk8@s.okayama-u.ac.jp (S.N.); mnakano@okayama-u.ac.jp (M.M.-N.); 5Department of Pediatric Dentistry, Graduate School of Dentistry, The University of Osaka, Suita 565-0871, Japan; suehiro.yuto.dent@osaka-u.ac.jp (Y.S.); nakano.kazuhiko.dent@osaka-u.ac.jp (K.N.); 6Division of Nephrology, Seirei Hamamatsu General Hospital, Hamamatsu 430-8558, Japan; misakitar@gmail.com; 7Department of Nursing, Faculty of Nursing, Seirei Christopher University, Hamamatsu 433-8558, Japan; 8Department of Internal Medicine, Japan Self-Defense Force Iruma Hospital, Iruma 358-0001, Japan; seigoemon@yahoo.co.jp

**Keywords:** red complex species, dental plaque, *Porphyromonas gingivalis*, *Tannerella forsythia*, oral microbiome

## Abstract

Red complex bacteria (*Porphyromonas gingivalis*, *Treponema denticola*, and *Tannerella forsythia*) have high virulence in periodontal disease. In the present study, we aimed to elucidate the detailed symbiotic relationships between the red complex and other oral bacteria in older Japanese individuals. Polymerase chain reaction was performed using dental plaque from 116 subjects and specific primers for ten periodontal pathogens. The detection rate of *Prevotella intermedia* and *Capnocytophaga sputigena* was significantly higher in *P. gingivalis*-positive subjects than in *P. gingivalis*-negative subjects (*p* < 0.05). The detection rate of *Campylobacter rectus*, *Prevotella nigrescens*, *Capnocytophaga ochracea*, and *Eikenella corrodens* was significantly higher in *T. forsythia*-positive subjects than in *T. forsythia*-negative subjects (*p* < 0.01). In a comprehensive analysis of oral microbiomes, three red complex species-positive subjects had significantly higher α-diversity than only *P. gingivalis*-positive subjects (*p* < 0.05) and had significantly lower β-diversity than only *T. forsythia*-positive subjects (*p* < 0.01). In the taxonomy analysis, *Porphyromonas* was significantly higher in three red complex species-positive subjects than in only *P. gingivalis*-positive and only *T. forsythia*-positive subjects (*p* < 0.01). These results suggest that each red complex species forms a unique oral microbiome and individuals positive for all red complex bacteria may harbor oral bacteria that confer a significant advantage in developing periodontal disease.

## 1. Introduction

The characteristics of biofilms formed by oral commensal bacteria are determined by an individual’s oral environment, including nutrients, temperature, pH, and anaerobicity [1,2]. A change in the oral environment from symbiosis to dysbiosis may cause a microbial shift that is problematic for humans [3]. Bacterial species within the biofilm are closely related to each other, and the increase in pathogenicity is due to a microbial shift of commensal oral bacteria rather than the entry of new bacterial species [4]. When the biofilm contains bacterial species with high periodontal pathogenicity, dysbiosis occurs as a result of microbial shift [3]. Periodontal disease develops as the bacterial species within the biofilm become highly pathogenic [3]. *Porphyromonas gingivalis* is well known as a keystone pathogen that induces microbial shift [5].

Recently, the strong relationship between periodontopathic bacteria and various systemic diseases such as IgA nephropathy, non-alcoholic fatty liver disease and cancer has become clear, and its importance is increasingly recognized [6,7,8,9,10,11,12,13]. Periodontal disease is caused by multiple causative bacteria and has a hierarchical structure based on pathogenicity [14,15]. The red complex bacteria (*P. gingivalis*, *Treponema denticola*, and *Tannerella forsythia*) have particularly high periodontal pathogenicity [16,17]. Orange complex bacteria including *Campylobacter rectus*, *Prevotella intermedia*, and *Prevotella nigrescens* are involved in the colonization of the red complex [18,19]. Less pathogenic commensal bacteria, such as green complex bacteria, are required for colonization of the orange complex [20,21]. As humans age, bacteria with low periodontal pathogenicity first begin to colonize in the human oral cavity, followed by the establishment of bacteria with higher periodontal pathogenicity [22]. However, the details of the symbiotic relationship between each oral bacterial species are unknown.

It has been established that *P. gingivalis*, *T. denticola*, and *T. forsythia* are at the top of the hierarchical structure of periodontopathic bacteria [14,15]. However, no studies have clarified which oral bacteria affect the colonization of each red complex species. In this study, we analyzed the association between each red complex species and other oral bacteria in older Japanese people using a conventional polymerase chain reaction (PCR) method and microbiome analysis.

## 2. Results

### 2.1. Detection of Major Periodontopathic Bacterial Species

The study enrolled 116 older Japanese subjects who did not require nursing care and had no serious underlying medical conditions (age range 61–91 years, median 75 years). The subject characteristics are shown in Table 1 and Appendix A Appendix A. There were no significant differences in the clinical characteristics between the groups used in this study.

Dental plaque was taken from the subjects and bacterial DNA was extracted from the dental plaque. PCR was performed using specific primers for ten major periodontal pathogens (*P. gingivalis*, *T. denticola*, *T. forsythia*, *Capnocytophaga ochracea*, *Capnocytophaga sputigena*, *P. intermedia*, *P. nigrescens*, *C. rectus*, *Aggregatibacter actinomycetemcomitans*, *Eikenella corrodens*). *P. gingivalis*, *T. forsythia*, *C. rectus*, *P. nigrescens*, *C. ochracea*, *C. sputigena*, and *E. corrodens* were detected in 60–90%, while *T. denticola*, *P. intermedia*, and *A. actinomycetemcomitans* were detected in 10–30% of subjects (Figure 1A). Regarding the relationship between periodontopathic bacteria, the detection rate of *P. intermedia* and *C. sputigena* was significantly higher in *P. gingivalis*-positive subjects than in *P. gingivalis*-negative subjects (*p* < 0.05) (Figure 1B, Table 2). There was no difference in the distribution of periodontopathic bacteria between *T. denticola*-positive subjects and *T. denticola*-negative subjects (Figure 1C), and the detection rate of *C. rectus*, *P. nigrescens*, *C. ochracea*, and *E. corrodens* was significantly higher in *T. forsythia*-positive subjects than in *T. forsythia*-negative subjects (*p* < 0.01) (Figure 1D, Table 3). The distribution of periodontopathic bacterial species according to the presence or absence of bacteria other than red complex species is shown in Appendix A.

The detection rate of *P. nigrescens*, *C. sputigena*, and *E. corrodens* was significantly higher in subjects who were positive for all three red complex species (*P. gingivalis*, *T. denticola*, and *T. forsythia*) than in subjects who were negative for all three red complex species (*p* < 0.05) (Figure 1E). Additionally, subjects who were positive for *P. gingivalis*, *T. denticola*, and *T. forsythia* had significantly more types of periodontopathic bacteria than subjects who were positive only for *P. gingivalis* or those who did not carry any of these bacteria (*p* < 0.05) (Figure 1F).

### 2.2. α-Diversity and β-Diversity

Next, microbiome analysis was performed by classifying the subjects into the following four groups, focusing on the presence or absence of red complex species: only positive for *P. gingivalis* (*Pg*-positive), only positive for *T. forsythia* (*Tf*-positive), positive for all red complex species (all red complex-positive), and positive for none of the red complex species (no red complex-positive). Only one subject was positive only for *T. denticola* and was excluded from the analysis. The mean operational taxonomic unit (OTU) number was significantly higher in the all red complex-positive group than the *Pg*-positive group (*p* < 0.05) (Figure 2A). Chao1 and Faith’s phylogenetic diversity analyses of α-diversity were highest in the all red complex-positive group and lowest in the *Pg*-positive group, with significant differences between the two groups (*p* < 0.05) (Figure 2B,C). For β-diversity, principal coordinate analysis of the unweighted UniFrac distance showed that the inter-sample distance was lowest in the all red complex-positive group and highest in the *Tf*-positive group, and there were significant intergroup differences between both groups (*p* < 0.01) (Figure 2D,E).

### 2.3. Taxonomic Analysis at the Phylum and Genus Levels

In the phylum level analysis, five phyla showed a relative abundance of more than 1%, accounting for more than 99% of the total. *Firmicutes* accounted for the largest proportion in many subjects (Figure 3A). The *Pg*-positive group and the *Tf*-positive group had several subjects with more *Actinobacteria* than *Firmicutes*, and each group except the all red complex-positive group had one subject with more *Proteobacteria* than *Firmicutes*. There was no significant difference in the relative ratio of phyla between each group (Figure 3B). In the genus level analysis, 14 genera in each group showed a relative abundance of 1% or more. Among these, the all red complex-positive group showed a significantly higher relative abundance of *Porphyromonas*, a member of the phylum *Bacteroidetes*, than the *Pg*-positive group and the *Tf*-positive group (Figure 4).

## 3. Discussion

In the present study, we analyzed the distribution of oral bacterial species related to the presence or absence of the red complex species (*P. gingivalis*, *T. denticola*, and *T. forsythia*) in older subjects. Detection of major periodontal pathogens by PCR revealed that *P. gingivalis* and *T. forsythia* coexisted with specific low pathogenic periodontal pathogens, whereas *T. denticola* had no such preferred coexistence. Additionally, subjects who were positive for all red complex species coexisted with more types of low pathogenic periodontopathic bacteria than subjects positive only for *P. gingivalis*. In the microbiome analysis, subjects who were positive for all red complex species had a higher proportion of the *Porphyromonas* genus and different oral microbiota diversity compared with subjects who were positive only for *P. gingivalis* and *T. forsythia*.

Among the ten major periodontopathic bacterial species, the detection frequencies of *P. gingivalis*, *T. forsythia*, *C. rectus*, *P. nigrescens*, *C. ochracea*, *C. sputigena*, and *E. corrodens* were approximately 60–80%, and the detection frequencies of *T. denticola*, *P. intermedia*, and *A. actinomycetemcomitans* were approximately 10–30%. This trend was similar to a previous report targeting Japanese people [23]. Interestingly, the presence or absence of *P. gingivalis* was significantly related to the presence or absence of *P. intermedia* and *C. sputigena*. In contrast, the presence or absence of *T. forsythia* was significantly related to the presence or absence of other bacteria such as *C. rectus*, *P. nigrescens*, *C. ochracea*, and *E. corrodens*. There were no periodontopathic bacteria related to the presence or absence of *T. denticola*. The results suggest that the composition of periodontopathic bacterial species in the biofilm may differ when *P. gingivalis* and *T. forsythia* are present.

When all red complex species were present, significantly more types of periodontopathic bacterial species were detected than when only *P. gingivalis* was present. Within the biofilm, various microbial interactions such as quorum sensing, coaggregation via cell surface adhesion, and metabolic cooperation can occur between bacterial species, supporting the growth of both species [24,25]. Microbiome analysis has confirmed a positive correlation between the quantities of multiple bacterial species in subgingival biofilm samples collected from periodontal patients [26]. Among the species of the red complex, mutualistic biofilm growth can occur, promoting cohabitation and proliferation [27,28,29]. For example, the metabolic products produced by *P. gingivalis* and *T. denticola* can each be used as nutritional substrates by the other [30]. Furthermore, co-culturing these species increases the expression of specific genes such as RgpA and gingipains in *P. gingivalis*, which promotes the adhesion to surface proteins of oral commensal bacteria [31]. Thus, periodontopathic bacteria in biofilms involve complex interactions with other coexisting commensal bacteria [32,33,34]. In the present study, the presence of all red complex species may create an environment with an increased variety of low pathogenic periodontopathic bacteria compared to when *P. gingivalis* exists alone.

Species α-diversity in each sample was analyzed via OTUs, Chao1, and Faith’s phylogenetic diversity. Comparing the number of OTUs detected in both groups revealed that the composition of bacterial species was significantly richer in the all red complex-positive group than in the *Pg*-positive group. Chao1 is a classical analysis based on the richness of observed species of α-diversity [35], and Faith’s phylogenetic diversity analysis shows evolutionary population diversity that considers the branch length of the phylogenetic tree based on the number of OTUs [35]. In both Chao1 and Faith’s phylogenetic diversity analyses, bacterial species diversity was higher in the all red complex-positive group than in the *Pg*-positive group. Using β-diversity, the degree of difference in diversity of each sample was expressed as the distance between each subject [36]. β-diversity was significantly lower in the all red complex-positive group than in the *Tf*-positive group, indicating that there were large differences in the composition of the oral microbiota between these groups. The results of α-diversity and β-diversity analysis revealed that when *P. gingivalis*, *T. denticola*, and *T. forsythia* are all positive, the microbial community structure diversity differs from when only *P. gingivalis* or *T. forsythia* is positive.

The red complex is a major group of bacteria involved in a microbial shift associated with the progression of periodontal disease [37,38]. Our PCR results showed that the type of major periodontopathic bacterial species was significantly higher in the all red complex-positive group than in the no red complex-positive group. In contrast, there was no significant difference in α-diversity or β-diversity between the all red complex-positive group and the no red complex-positive group. Therefore, bacterial diversity in the entire oral microbiome of all red complex-positive subjects did not significantly differ from that of no red complex-positive subjects, and the red complex shapes a unique oral microbiome specific for periodontal pathogens, which may be enhanced by the presence of all red complex species.

In the phylum-level analysis, five phyla (*Proteobacteria*, *Firmicutes*, *Actinobacteria*, *Bacteroidetes*, and *Fusobacteria*) were the most prevalent, with relative abundances above 1%. There was no significant difference in the relative ratio of these five phyla depending on the presence or absence of *P. gingivalis*, *T. denticola*, and *T. forsythia*. In the present study targeting older Japanese people, only Firmicutes among the five phyla showed an extremely high relative ratio, whereas in a previous study targeting Japanese children, three phyla—*Firmicutes*, *Proteobacteria*, and *Actinobacteria*—showed high relative ratios, and the relative ratios of *Bacteroidetes* and *Fusobacteria* were not as low as those in the present study [39]. Therefore, with age, phyla variation may decrease and the proportion of *Firmicutes* may increase. Furthermore, because the oral health status of children still varies greatly their systemic condition [40,41], further studies on the oral microbiota should be conducted, focusing on differences in various backgrounds.

At the genus level analysis, 14 genera (*Actinomyces*, *Corynebacterium*, *Rothia*, *Porphyromonas*, *Prevotella*, *Streptococcus*, *Veillonella*, *Neisseria*, *Haemophilus*, *Pseudomonas*, *Capnocytophaga*, *Granulicatella*, *Fusobacterium*, and *Leptotrichia*) were the most prevalent in all groups, with relative abundance of more than 1%. There was no significant difference in the relative ratio of 13 out of 14 genera between the presence and absence of *P. gingivalis*, *T. denticola*, and *T. forsythia*. *P. gingivalis* causes microbial shift and dysbiosis of the oral flora [5], while our results show that *P. gingivalis*-induced microbial shift does not occur on a large scale at the phyla and genus levels.

Interestingly, the relative abundance of the *Porphyromonas* genus was significantly higher in the all red complex-positive group than in the *Pg*-positive group and the *Tf*-positive group. Periodontitis is caused by hundreds of different bacterial species present in the biofilm accumulated periodontal tissues, and it is influenced by the complex interactions between the bacteria and the host [42,43]. The biofilm bacteria and their byproducts stimulate the gingival epithelium, leading to an inflammatory response [44]. *P. gingivalis* may disrupt the host’s response in periodontal tissues, potentially causing harmful effects [44]. The biofilm community structure differs depending on the presence and absence of *P. gingivalis* [44]. Interestingly, experiments using germ-free mice have shown that *P. gingivalis* induces inflammation in a polymicrobial community and does not cause disease when present alone [45]. Taken together, the presence of all red complex species in the microbiota may favor the survival of *P. gingivalis* within the *Porphyromonas* genus, potentially influencing its quantitative proliferation.

This study has some limitations. First, it was conducted as part of a comprehensive health assessment that included evaluations of physical function, mental state, nutritional status, and social environment in older adults. As a result, we could not obtain information regarding periodontal status and brushing habits. Additionally, we attempted to collect information on medication usage. However, we encountered challenges such as participants not being aware of their medication regimens, lacking a medication list, and discrepancies between self-reported medication and the actual medication list, which hindered our ability to obtain reliable information. Secondly, there was no significant correlation between the types of bacteria and health status. This is likely because the project aimed to detect frailty early to prevent the onset of severe diseases, and the subjects were relatively healthy older adults who could come to the testing site on their own. Therefore, it can be assumed that no correlation between the types of bacteria and health status was found. However, to our knowledge, this is the first study to clarify the relationship between each species of the red complex and other oral bacterial species. Thus, we believed obtaining primary data using older adults without underlying severe conditions or systemic effects was meaningful. We must conduct longitudinal prospective studies or research targeting individuals with various systemic diseases who require caregiving to compare our current results with these future findings.

Recent molecular analyses of oral bacterial species, including periodontal pathogens, have yielded significant insights related to health [46]. While the review article included new findings, most studies had a small sample size, ranging from 10 to several dozen subjects. Nevertheless, systematic reviews and meta-analyses based on these manuscripts have shown promising associations [46]. Based on these studies, we decided to conduct research with a larger sample size. We collected sequential samples for all participants during four cohorts for one year to reduce bias.

A schematic diagram of this study is shown in Figure 5. The abundance of orange complex and green complex species in dental plaque varied depending on the presence or absence of red complex species; *Pg*-positive and *Tf*-positive subjects showed predominant coexistence of specific orange complex and green complex species, respectively, but such coexistence with orange and green complex species was not prevalent in *Td*-positive and no red complex-positive subjects. All red complex positive-subjects had higher α-diversity than only *Pg*-positive subjects and higher β-diversity than only *Tf*-positive subjects. Furthermore, all red complex-positive subjects had a higher abundance of the *Porphyromonas* genus than *Pg*-positive and *Tf*-positive subjects. These results suggest that the composition of oral bacteria including periodontopathic bacteria varies significantly depending on the distribution of each red complex species in the oral cavity.

## 4. Materials and Methods

### 4.1. Ethics Statement

The study protocol was approved by the Human Genome and Genetic Analysis Research Ethics Review Committee of Hyogo Medical University (Approval number: 0342). Prior to specimen collection, all subjects were informed of the study protocol and provided written informed consent.

### 4.2. Subjects and Specimens

This study was designed as part of the Frail Elderly in the Sasayama-Tamba Area (FESTA) study [47,48]. A total of 116 subjects (39 men, 77 women; age range 64–91 years, median 75 years, mean age 75.5 ± 5.3 years) who lived in the Tamba-Sasayama region of Hyogo Prefecture, Japan, were enrolled (Table 1). These subjects required no nursing care and had no serious underlying diseases. From October 2022 to December 2022, dental plaque was collected using seed swabs from whole tooth surfaces of the subjects. The samples were suspended in 1 mL of phosphate-buffered saline and centrifuged, and the supernatant was discarded. The pellet was immediately used for the microbiological analysis described below.

### 4.3. DNA Extraction

Bacterial DNA was extracted from swab specimens. Briefly, a swab specimen was resuspended in 250 μL of 10 mM Tris-HCl (pH 8.0) containing 100 mM NaCl and 1 mM EDTA (Sigma-Aldrich, St. Louis, MO, USA). The cells were collected using centrifugation and lysed in 600 µL of cell lysis solution (Qiagen, Düsseldorf, Germany) and incubated at 80 °C for 5 min, followed by the addition of 3 μL of RNase A (10 mg/mL; Qiagen) and incubation at 37 °C for 30 min. Protein precipitation solution (Qiagen) was added, vigorously vortexed for 20 s, and then centrifuged at 10,000× *g* for 3 min. The supernatant was combined with 600 μL of isopropanol (Fujifilm Wako Pure Chemical Corporation, Osaka, Japan) and centrifuged. The precipitate was then resuspended in 70.0% ethanol (Fujifilm Wako Pure Chemical Corporation), centrifuged, combined with 100 μL of sterile distilled water, and used in the following study.

### 4.4. PCR Detection of Periodontopathic Bacterial Species

PCR was performed using bacterial DNA extracted from the oral specimens to detect ten major periodontopathic bacterial species (*P. gingivalis*, *T. denticola*, *T. forsythia*, *C. ochracea*, *C. sputigena*, *P. intermedia*, *P. nigrescens*, *C. rectus*, *A. actinomycetemcomitans*, *E. corrodens*) with species-specific sets of primers (Greiner Bio-One, Kremsmünster, Austria) and TaKaRa Ex *Taq* polymerase (Takara Bio. Inc., Otsu, Japan) [49,50,51,52,53,54]. (Table 4). PCR amplification using each primer set was performed using a thermal cycler (Bio-rad Laboratories Inc., Hercules, CA, USA) with the following cycling parameters, as described previously [23]: initial denaturation at 95 °C for 4 min; followed by 30 cycles at 95 °C for 30 s, 58 °C for 30 s, and 72 °C for 30 s; with a final extension at 72 °C for 7 min. The PCR products were fractionated in a 1.5% (*w*/*v*) agarose gel-Tris-acetate-EDTA buffer (Nippon Gene Co., Ltd., Tokyo, Japan), then stained with ethidium bromide (0.5 μg/mL) (Nacalai Tesque Inc., Kyoto, Japan) and visualized under UV light using FAS-V (Nippon Genetics Co., Ltd., Tokyo, Japan).

### 4.5. 16S rRNA Gene Library Preparation, Sequencing, and OTU Analysis

Microbiome analyses were performed according to the previously described methods [39]. For 16S rRNA gene sequencing, bacterial DNA from dental plaque specimens were used. Bioinformatic analyses were conducted using quantitative insights into microbial ecology 2 (Qiime 2). The quality of the reads obtained from all specimens were evaluated, and only reads considered to be appropriate were determined. An OTU table was then made using Divisive Amplicon Denoising Algorithm 2 (Dada2). Thereafter, diversity analysis and taxonomic classification were performed. The phylum was extracted from taxa for which mean relative abundance was more than 1%. The genus was extracted where OTUs were present in more than 50% of the specimens, according to the method previously described [39].

### 4.6. Statistical Analysis

Statistical analyses were performed using GraphPad Prism 9 (GraphPad Software Inc., La Jolla, CA, USA) and STATA version 15 (StataCorp LLC, College Station, TX, CA, USA). Comparisons between two groups (bacteria-positive and bacteria-negative specimens) were performed using Fisher’s exact probability test, as well as multivariate logistic regression analysis models. For comparisons between groups of periodontopathic bacteria, α-diversity, β-diversity, and taxonomic analysis, the Kruskal–Wallis test was used for nonparametric analysis, followed by the Dunn test for multiple comparisons. Age adjustments were made between groups of different ages. Differences were considered statistically significant at *p* < 0.05.

## 5. Conclusions

The distribution of periodontopathic bacterial species was different among subjects who were positive for each of *P. gingivalis*, *T. denticola*, and *T. forsythia*. All red complex-positive subjects had a wide variety of periodontal disease-related bacterial species. The oral microbiome of all red complex-positive subjects contained a large amount of *Porphyromonas* genus. Our results suggest that a unique microbiome is formed in the oral cavity of subjects with each red complex, which may influence oral and systemic health.

## Figures and Tables

**Figure 1 ijms-25-12243-f001:**
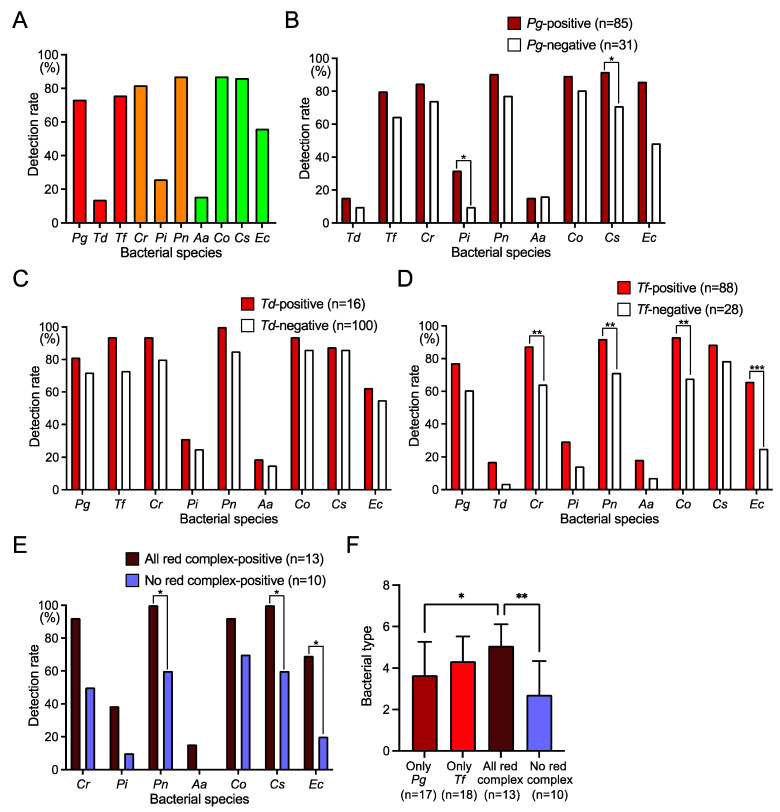
Distribution of periodontopathic bacterial species in dental plaque specimens. (**A**) Detection rates of each periodontopathic bacterial species. (**B**–**E**) Distribution of each periodontopathic bacterial species in (**B**) *Porphyromonas gingivalis*-positive and *P. gingivalis*-negative subjects, (**C**) *Treponema denticola*-positive and *T. denticola*-negative subjects, (**D**) *Tannerella forsythia*-positive and *T. forsythia*-negative subjects, (**E**) all red complex-positive and no red complex-positive subjects. (**F**) Bacterial type of periodontopathic bacterial species other than red complex. * *p* < 0.05, ** *p* < 0.01, *** *p* < 0.001 between each group. *Pg*, *P. gingivalis*; *Td*, *T. denticola*; *Tf*, *T. forsythia*; *Cr*, *Campylobacter rectus*; *Pi*, *Prevotella intermedia*; *Pn*, *Prevotella nigrescens*; *Aa*, *Aggregatibacter actinomycetemcomitans*; *Co*, *Capnocytophaga ochracea*; *Cs*, *Capnocytophaga sputigena*; and *Ec*, *Eikenella corrodens*.

**Figure 2 ijms-25-12243-f002:**
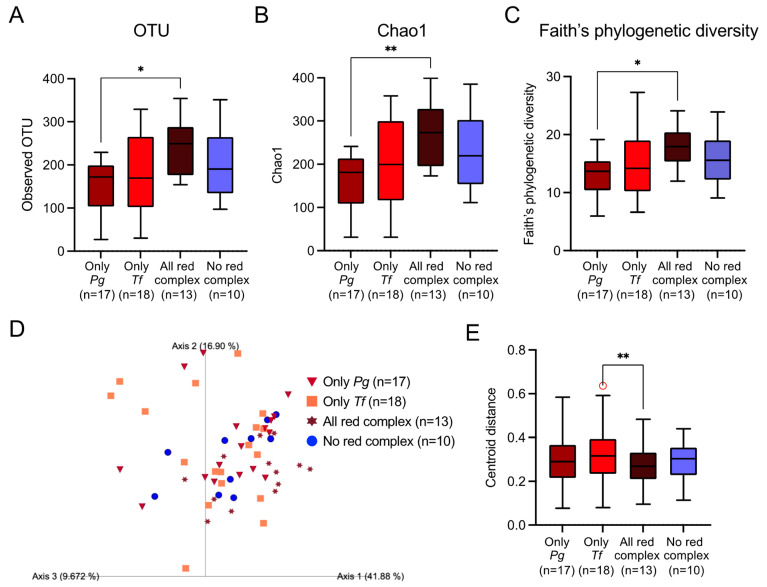
α-Diversity and β-diversity in dental plaque specimens. (**A**–**C**) Analyses of α-diversity. (**A**) Number of operational taxonomic units (OTUs), (**B**) Chao1, (**C**) Faith’s phylogenetic diversity. Whiskers indicate maximum and minimum values, boxes indicate the interquartile range, horizontal lines within boxes indicate median values. (**D**,**E**) Analyses of β-diversity. (**D**) Weighted UniFrac distance. The contribution rate of each Axis (1–3) in the principal coordinate analysis is shown in parentheses. In this analysis, Axis 1 (41.8%) + Axis 2 (16.9%) + Axis 3 (9.7%) = 68.4%, which is considered to reflect approximately 70% of all the information. (**E**) PERMANOVA analysis of the weighted UniFrac distance. Whiskers indicate maximum and minimum values, boxes indicate the interquartile range, horizontal lines within boxes indicate median values, and circles indicate outliers. * *p* < 0.05 and ** *p* < 0.01 between each group. *Pg*, *Porphyromonas gingivalis*; *Tf*, *Tannerella forsythia*.

**Figure 3 ijms-25-12243-f003:**
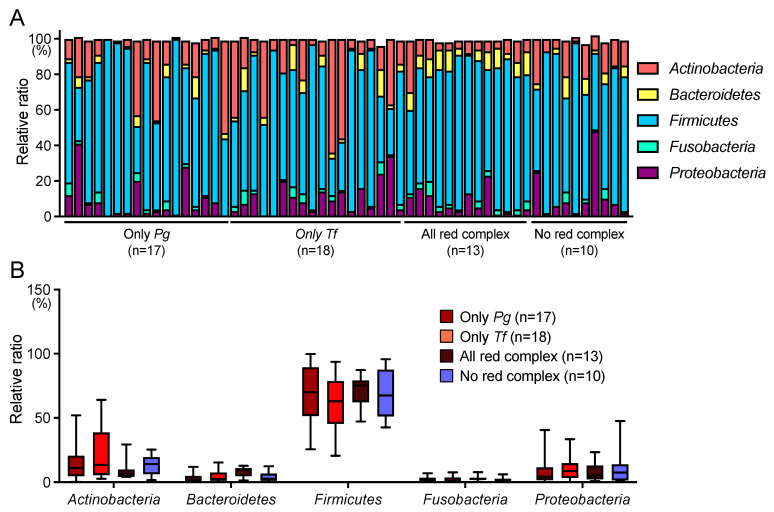
Taxonomic analysis at the phylum level. (**A**) Relative abundance of oral microbiota for each subject. (**B**) Relative abundance of six major phyla. Whiskers indicate maximum and minimum values, boxes indicate the interquartile range, horizontal lines within boxes indicate median values. *Pg*, *Porphyromonas gingivalis*; *Tf*, *Tannerella forsythia*.

**Figure 4 ijms-25-12243-f004:**
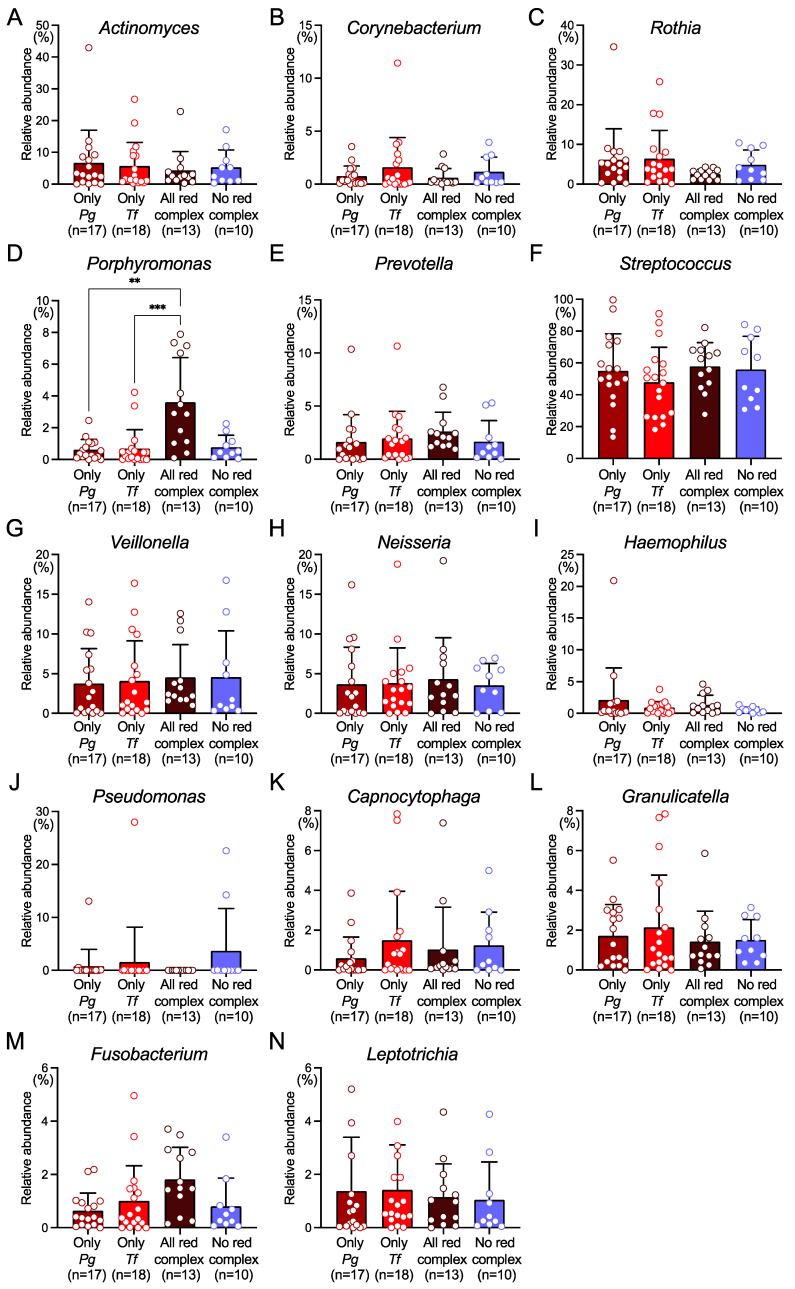
Taxonomic analysis at the genus level. Relative abundance of (**A**) *Actinomyces*, (**B**) *Corynebacterium*, (**C**) *Rothia*, (**D**) *Porphyromonas*, (**E**) *Prevotella*, (**F**) *Streptococcus*, (**G**) *Veillonella*, (**H**) *Neisseria*, (**I**) *Haemophilus*, (**J**) *Pseudomonas*, (**K**) *Capnocytophaga*, (**L**) *Granulicatella*, (**M**) *Fusobacterium*, (**N**) *Leptotrichia*. Data expressed as the mean ± standard deviation and circles indicate each subject. Each circle represents the data of each subject. ** *p* < 0.01 and *** *p* < 0.001 between each group. *Pg*, *Porphyromonas gingivalis*; *Tf*, *Tannerella forsythia*.

**Figure 5 ijms-25-12243-f005:**
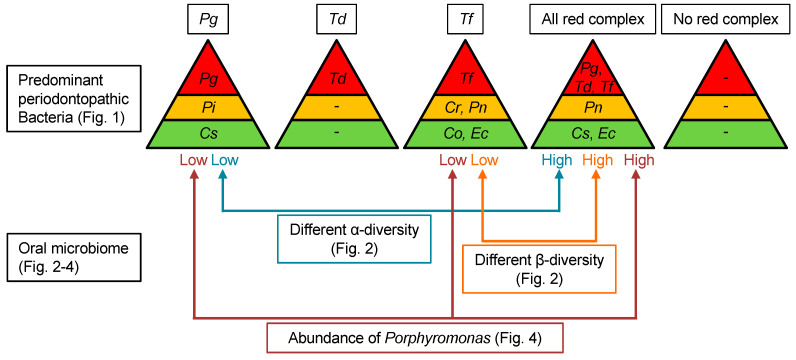
Schematic representation of the relationship between red complex and other oral bacteria. Pg, *Porphyromonas gingivalis*; Td, *Treponema denticola*; Tf, *Tannerella forsythia*; Cr, *Campylobacter rectus*; Pi, *Prevotella intermedia*; Pn, *Prevotella nigrescens*; Co, *Capnocytophaga ochracea*; Cs, *Capnocytophaga sputigena*; and Ec, *Eikenella corrodens*.

**Table 1 ijms-25-12243-t001:** Subjects’ characteristics.

Characteristics	*Pg*-Positive (n = 85)	*Td*-Positive	*Tf*-Positive	All Red Complex-Positive	No Red Complex-Positive	Total
		(n = 16)	(n = 88)	(n = 13)	(n = 10)	(n = 116)
Age (year)	75.5 ± 5.3	75.9 ± 5.7	75.4 ± 5.0	76.2 ± 6.2	74.7 ± 5.8	75.5 ± 5.3
Sex (male:female)	27:58	6:10	26:62	4:9	5:5	39:77
Height (cm)	154.7 ± 8.1	155.7 ± 5.6	154.8 ± 8.2	155.9 ± 5.8	157.7 ± 8.2	155.0 ± 8.2
Weight (cm)	53.6 ± 8.7	54.7 ± 8.7	53.7 ± 8.5	55.2 ± 8.5	55.8 ± 12.9	53.7 ± 8.9
Systolic blood pressure (mmHg)	144.6 ± 18.4	135.3 ± 12.6	145.1 ± 17.7	136.0 ± 13.2	144.3 ± 13.8	144.1 ± 18.4
Diastolic blood pressure (mmHg)	82.4 ± 11.6	77.6 ± 9.8	83.1 ± 12.2	77.0 ± 9.7	79.5 ± 9.2	82.3 ± 11.8
Hypertension (%)	18 (21.2%)	1 (6.3%)	19 (21.6%)	1 (7.7%)	1 (10.0%)	23 (19.8%)
Diabetes mellitus (%)	0 (0.0%)	0 (0.0%)	0 (0.0%)	0 (0.0%)	0 (0.0%)	0 (0.0%)
Dyslipidemia (%)	6 (7.1%)	0 (0.0%)	6 (6.8%)	0 (0.0%)	0 (0.0%)	8 (6.9%)
Liver disease (%)	2 (2.4%)	0 (0.0%)	2 (2.3%)	0 (0.0%)	0 (0.0%)	2 (1.7%)
Chronic kidney disease (%)	0 (0.0%)	0 (0.0%)	1 (1.1%)	0 (0.0%)	0 (0.0%)	1 (0.9%)
Heart disease (%)	4 (4.7%)	1 (6.3%)	3 (3.4%)	1 (7.7%)	1 (10.0%)	5 (4.3%)
Asthma (%)	0 (0.0%)	0 (0.0%)	1 (1.1%)	0 (0.0%)	0 (0.0%)	1 (0.9%)
Tuberculosis (%)	0 (0.0%)	0 (0.0%)	0 (0.0%)	0 (0.0%)	0 (0.0%)	0 (0.0%)
Lung infection (%)	0 (0.0%)	0 (0.0%)	0 (0.0%)	0 (0.0%)	0 (0.0%)	0 (0.0%)
Digestive disease (%)	1 (1.2%)	0 (0.0%)	1 (1.1%)	0 (0.0%)	0 (0.0%)	1 (0.9%)
Osteoporosis (%)	2 (2.4%)	0 (0.0%)	2 (2.3%)	0 (0.0%)	0 (0.0%)	2 (1.7%)
Rheumatism (%)	0 (0.0%)	0 (0.0%)	0 (0.0%)	0 (0.0%)	0 (0.0%)	0 (0.0%)
Thyroid disease (%)	3 (3.5%)	0 (0.0%)	4 (4.5%)	0 (0.0%)	1 (10.0%)	5 (4.3%)
Collagen disease (%)	1 (1.2%)	0 (0.0%)	1 (1.1%)	0 (0.0%)	0 (0.0%)	1 (0.9%)
Blood disease (%)	2 (2.4%)	0 (0.0%)	1 (1.1%)	0 (0.0%)	0 (0.0%)	2 (1.7%)
Stroke (%)	0 (0.0%)	0 (0.0%)	0 (0.0%)	0 (0.0%)	0 (0.0%)	0 (0.0%)
Cancer (%)	4 (4.7%)	0 (0.0%)	3 (3.4%)	0 (0.0%)	0 (0.0%)	5 (4.3%)
White blood cell (10^2^/μL)	56.7 ± 14.5	64.3 ± 17.2	56.7 ± 13.5	63.5 ± 18.8	62.2 ± 10.2	57.4 ± 13.6
Red blood cell (10^4^/μL)	441.7 ± 38.7	443.9 ± 31.0	447.3 ± 38.6	437.5 ± 28.7	453.9 ± 57.2	445.2 ± 40.9
Hemoglobin (g/dL)	13.2 ± 1.1	13.4 ± 1.0	13.4 ± 1.0	13.1 ± 1.1	13.1 ± 1.0	13.3 ± 1.1
Hematocrit (%)	42.0 ± 3.3	42.2 ± 3.2	42.5 ± 3.2	41.8 ± 3.1	41.8 ± 3.0	42.2 ± 3.3
Platelet (10^4^/μL)	24.3 ± 4.8	23.9 ± 5.3	24.5 ± 4.9	24.3 ± 4.4	23.0 ± 6.2	24.2 ± 5.0
HDL cholesterol (mg/dL)	65.4 ± 14.9	69.8 ± 19.4	67.4 ± 14.4	65.9 ± 14.9	66.8 ± 16.0	65.6 ± 14.6
Total protein (g/dL)	7.2 ± 0.4	7.3 ± 0.4	7.2 ± 0.4	7.2 ± 0.4	7.1 ± 0.3	7.2 ± 0.4
Blood sugar (mg/dL)	111.4 ± 34.3	115.0 ± 23.7	112.2 ± 34.6	110.8 ± 36.9	117.2 ± 42.4	112.7 ± 34.0
Total cholesterol (mg/dL)	210.0 ± 36.5	212.4 ± 51.7	209.1 ± 37.5	209.6 ± 35.5	203.9 ± 38.6	207.3 ± 36.7
LDL cholesterol (mg/dL)	116.1 ± 28.9	116.3 ± 38.9	112.6 ± 27.7	114.9 ± 28.1	112.5 ± 31.3	113.3 ± 28.4
Triglyceride (mg/dL)	142.5 ± 75.3	131.3 ± 67.0	145.3 ± 74.6	143.8 ± 79.1	123.1 ± 70.6	142.0 ± 72.0
Creatinine (mg/dL)	0.7 ± 0.2	0.7 ± 0.2	0.7 ± 0.2	0.7 ± 0.2	0.8 ± 0.2	0.7 ± 0.2
γ-GTP (U/I)	35.1 ± 84.3	125.8 ± 249.6	34.0 ± 82.8	37.0 ± 93.5	32.6 ± 26.1	38.6 ± 97.9
AST (U/L)	25.8 ± 9.2	31.4 ± 17.0	25.6 ± 8.2	26.4 ± 9.6	23.9 ± 11.8	25.9 ± 9.5
ALT (U/L)	20.9 ± 8.2	25.3 ± 11.9	21.2 ± 7.6	21.3 ± 8.4	16.9 ± 7.2	20.8 ± 7.8
Albumin (g/dL)	4.2 ± 0.2	4.2 ± 0.2	4.3 ± 0.2	4.3 ± 0.2	4.2 ± 0.3	4.2 ± 0.3
Cystatin C (mg/L)	1.0 ± 0.3	1.0 ± 0.2	1.0 ± 0.2	1.0 ± 0.2	1.0 ± 0.1	1.0 ± 0.2
High sensitive CRP (mg/dL)	0.1 ± 0.1	0.1 ± 0.1	0.1 ± 0.1	0.1 ± 0.1	0.1 ± 0.2	0.1 ± 0.1
Teeth number	21.2 ± 7.0	23.8 ± 5.6	21.6 ± 7.3	23.2 ± 5.8	20.8 ± 6.1	21.3 ± 7.2

Age, height, weight, blood pressure, blood test data, and teeth number are expressed as mean ± standard deviation. *Pg*, *Porphyromonas gingivalis*; *Td*, *Treponema denticola*; *Tf*, *Tannerella forsythia*.

**Table 2 ijms-25-12243-t002:** Multivariate analysis in the presence or absence of *Porphyromonas gingivalis*.

	Model 1				Model 2	
Variables	Odds Ratio (95% Confidence Interval)	*p* Value	Odds Ratio (95% Confidence Interval)	*p* Value	Odds Ratio (95% Confidence Interval)	*p* Value
Age	1.00 (0.91–1.09)	0.955	1.01 (0.92–1.20)	0.901	0.99 (0.91–1.09)	0.856
Sex	1.48 (0.60–3.63)	0.398	1.57 (0.63–3.90)	0.330	1.63 (0.64–3.14)	0.306
Teeth number	0.98 (0.92–1.05)	0.596	0.99 (0.92–1.05)	0.691	0.98 (0.91–1.04)	0.485
*Pi*	**4.22 (1.16–15.4)**	**0.029**	-	-	3.56 (0.95–13.3)	0.059
*Cs*	-	-	**4.91 (1.60–15.0)**	**0.005**	**4.07 (1.30–12.8)**	**0.016**

Bold values indicate statistical significance at *p* < 0.05. *Pi*, *Prevotella intermedia*; *Cs*, *Capnocytophaga sputigena*.

**Table 3 ijms-25-12243-t003:** Multivariate analysis in the presence or absence of *Tannerella forsythia*.

	Model 1								Model 2	
Variables	Odds Ratio (95% Confidence Interval)	*p* Value	Odds Ratio (95% Confidence Interval)	*p* Value	Odds Ratio (95% Confidence Interval)	*p* Value	Odds Ratio (95% Confidence Interval)	*p* Value	Odds Ratio (95% Confidence Interval)	*p* Value
Age	1.02 (0.93–1.12)	0.684	1.01 (0.93–1.11)	0.697	1.06 (0.96–1.16)	0.263	1.00 (0.91–1.10)	0.960	1.02 (0.93–1.13)	0.647
Sex	2.12 (0.83–5.33)	0.110	2.09 (0.82–5.28)	0.119	2.26 (0.87–5.89)	0.094	1.52 (0.59–3.96)	0.384	1.65 (0.59–4.59)	0.336
Teeth number	1.02 (0.95–1.09)	0.597	1.03 (0.97–1.10)	0.362	1.05 (0.98–1.12)	0.166	1.03 (0.97–1.10)	0.357	1.04 (0.97–1.12)	0.253
*Cr*	**3.74 (1.35–10.35)**	**0.011**	-	-	-	-	-	-	1.49 (0.44-5.09)	0.526
*Pn*	-	-	**4.68 (1.48–14.8)**	**0.009**	-	-	-	-	2.19 (0.53–9.04)	0.280
*Co*	-	-	-	-	**8.05 (2.26–27.50)**	**0.001**	-	-	**5.03 (1.29–19.6)**	**0.020**
*Ec*	-	-	-	-	-	-	**5.57 (2.06–15.2)**	**0.001**	**4.04 (1.41–11.6)**	**0.010**

Bold values indicate statistical significance at *p* < 0.05. *Cr*, *Campylobacter rectus*; *Pn*, *Prevotella nigrescens*; *Co*, *Capnocytophaga ochracea*; *Ec*, *Eikenella corrodens*.

**Table 4 ijms-25-12243-t004:** Polymerase chain reaction primers used in this study.

Purpose	Sequence (5′-3′)	Size (bp)	References
Universal primer			
(positive control)			
PA	AGA GTT TGA TCC TGG CTC AG	315	[49]
PD	GTA TTA CCG CGG CTG CTG		
Detection of periodontitis-related species			
*Porphyromonas gingivalis*	CCG CAT ACA CTT GTA TTA TTG CAT GAT A	267	[50]
	AAG AAG TTT ACA ATC CTT AGG ACT GTC T		
*Treponema denticola*	AAG GCG GTA GAG CCG CTC A	311	[51]
	AGC CGC TGT CGA AAA GCC CA		
*Tannerella forsythia*	GCG TAT GTA ACC TGC CCG CA	641	[52]
	TGC TTC AGT GTC AGT TAT ACC T		
*Capnocytophaga ochracea*	AGA GTT TGA TCC TGG CTC AG	185	[53]
	GAT GCC GTC CCT ATA TAC TAT GGG G		
*Capnocytophaga sputigena*	AGA GTT TGA TCC TGG CTC AG	185	[53]
	GAT GCC GCT CCT ATA TAC CAT TAG G		
*Prevotella intermedia*	TTT GTT GGG GAG TAA AGC GGG	575	[53]
	TCA ACA TCT CTG TAT CCT GCG T		
*Prevotella nigrescens*	ATG AAA CAA AGG TTT TCC GGT AAG	804	[52]
	CCC ACG TCT CTG TGG GCT GCG A		
*Campylobacter rectus*	TTT CGG AGC GTA AAC TCC TTT TC	598	[52]
	TTT CTG CAA GCA GAC ACT CTT		
*Aggregatibacter actinomycetemcomitans*	CTA GGT ATT GCG AAA CAA TTT G	262	[54]
	CCT GAA ATT AAG CTG GTA ATC		
*Eikenella corrodens*	CTA ATA CCG CAT ACG TCC TAA G	688	[52]
	CTA CTA AGC AAT CAA GTT GCC C		

## Data Availability

The data are available from the corresponding author upon reasonable request.

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
