# Peer review of "Relationship Between the Presence of Red Complex Species and the Distribution of Other Oral Bacteria, Including Major Periodontal Pathogens in Older Japanese Individuals"

_ijms, 2024, doi:10.3390/ijms252212243_

Round 1

Reviewer 1 Report

Comments and Suggestions for Authors

This is an interesting study. The authors present the data well and the results reflect the data. The only drawback is the “Discussion” part. Feels like the author present the results again, it would be better to add more discussion about why subjects who were positive for all red complex species coexisted with more types of low pathogenic periodontopathic bacteria than subjects positive only for P.gingivalis (bacteria competition, interaction…), why subjects who were positive for all red complex species had a higher proportion of the Porphyromonas genus (trigger inflammation…). I like the schematic diagram part. The resolution of all the tables have to be improved.

Minor comments:

1.      Figure 1F, it would be better to change Bacterial number to “Bacterial type”, otherwise people would think about the bacteria count. Also please change the bacterial number to bacterial type in the manuscript.

2.      Figure 2 lengend is not correct.

3.      Figure 4 only present 10 genus, not 14.

4.      Line 183, I suggest to remove “whose oral microbiota was already in a steady state”, oral microbiota will change based on the health condition, habit, environment, hormone, age…, it is hard to tell whether the old peoples oral microbiota is already in a steady state or not.

5.      Line 202, I suggest to remove “T.denticola”, because the author’s data already showed “there were no periodontopathic bacteria related to the presence or absence of T.denticola”.

Comments on the Quality of English Language

Multiple short sentences could be used to replace the long sentence. As a non-native English speaker, a long sentence might be challenged.

Reviewer 2 Report

Comments and Suggestions for Authors

The authors explored the presence of red complex bacteria in relation to other species in an older population in Japan. I have some recommendations and questions, as follows.

Regarding the methods, did the authors collect information regarding the periodontal status of the individuals included (e.g., diagnosis and level of periodontitis or gingivitis). Also, it would be important to present data such as brushing habits and the use of any medication.

Also, the authors stated that the volunteers did not present serious underlying medical conditions. However, there are chronic conditions such as diabetes, which could interfere with the oral microbiome, especially in an elder population. Please could you clarify wether you collect the information on the health status of the individuals included?

Please clarify why the authors chose to include 116 individuals. A convenience sample?

In the Discussion, the authors should outline the study limitations properly.

Please improve the quality of the figures which are not satisfactory.

Round 2

Reviewer 2 Report

Comments and Suggestions for Authors

I have no further concerns.